# Investigation of the Relationships between Coat Colour, Sex, and Morphological Characteristics in Donkeys Using Data Mining Algorithms

**DOI:** 10.3390/ani13142366

**Published:** 2023-07-20

**Authors:** Şenol Çelik, Orhan Yılmaz

**Affiliations:** 1Biometry and Genetic Unit, Department of Animal Science, Faculty of Agriculture, Bingol University, Bingol 12000, Turkey; 2Plant and Animal Production Department, Posof Vocational School, Ardahan University, Ardahan 75000, Turkey; orhanyilmaz@ardahan.edu.tr

**Keywords:** random forest, CHAID, MARS algorithm, morphological measurements

## Abstract

**Simple Summary:**

The donkey (*Equus asinus*) is an odd-toed ungulate and the smallest species in the Equidae family. It is characteristically short-legged with extremely long ears. The wild ancestor of the donkey is equally *Equus asinus*, which is generally known as the “African wild ass” and is reportedly still extant. Donkeys are the only ungual animal domesticated exclusively in Africa. By nature, donkeys are very companionable, calm, enduring, intelligent, prudent, playful, and keen to learn, and they enjoy the company of humans. In Turkey, donkeys are used for pack transport and riding in order to lessen the physical load on humans. This study was conducted to assess the prediction performance of various algorithms using the morphological traits, body coat colour distribution, and body measurements of donkeys raised in Turkey.

**Abstract:**

This study was carried out in order to determine the morphological characteristics, body coat colour distribution, and body dimensions of donkeys raised in Turkey, as well as to determine the relationships between these factors. For this reason, the predictive performance of various machine learning algorithms (i.e., CHAID, Random Forest, ALM, MARS, and Bagging MARS) were compared, utilising the biometric data of donkeys. In particular, mean measurements were taken from a total of 371 donkeys (252 male and 119 female) with descriptive statistical values as follows: height at withers, 100.7 cm; rump height, 103.1 cm; body length, 103.8 cm; chest circumference, 112.8 cm; chest depth, 45.7 cm; chest width, 29.1 cm; front shin circumference, 13.5 cm; head length, 55 cm; and ear length, 22 cm. The body colour distribution of the donkeys considered in this study was calculated as 39.35% grey, 19.95% white, 21.83% black, and 18.87% brown. Model fit statistics, including the coefficient of determination (R^2^), mean square error, root-mean-square error (RMSE), mean absolute percentage error (MAPE), and standard deviation ratio (SD ratio), were calculated to measure the predictive ability of the fitted models. The MARS algorithm was found to be the best model for defining the body length of donkeys, with the highest R^2^ value (0.916) and the lowest RMSE, MAPE, and SD ratio values (2.173, 1.615, and 0.291, respectively). The experimental results indicate that the most suitable model is the MARS algorithm, which provides a good alternative to other data mining algorithms for predicting the body length of donkeys.

## 1. Introduction

Donkeys have played a very important role in agricultural practices until recently, and they are still used for transportation in rural areas of Turkey, as well as in other countries where cultivation is still carried out [1,2]. They have traditionally been used as a beast of burden and, even though the world has moved toward mechanisation, this ancient animal is still being used as a biological vehicle. In dry and semi-dry areas, they serve humans by carrying luggage and are used for transportation purposes. Donkeys are also used to guard sheep, as they are more inclined to stand and fight than to run from a predator [3].

The donkey (*Equus africanus asinus*) belongs to the Equidae family, alongside horses, and has been domesticated in most parts of the world [4]. The lifespan of a donkey generally ranges from 25 to 50 years [3]. By nature, donkeys are very friendly, calm, quiet, patient, intelligent, cautious, playful, and eager to learn, and they enjoy the company of humans. They have great disease resistance and are both agile and cautious on bad terrain. Donkeys are hardy and tend to live longer than other species under the same conditions [5,6]. Donkeys have a lower water requirement per unit weight than other domesticated animals, aside from camels [7].

There are 140 indigenous donkey breeds worldwide, with most breeds originating from Asia, Europe, and the Caucasus [8]. The ancestors of the domestic donkey (*Equus asinus asinus*) are reported to have inhabited deserts in Sudan, Ethiopia, and Somalia. Characteristically, the withers height is 110–122 cm in the Nubian ass and 130–140 cm in the Somali ass [9].

In Turkey, donkeys are used for pack transport and riding in order to ease the physical burden on humans, carrying firewood, water, grain, hay, and goods over short distances. They escort sheep flocks and carry goods for shepherds, and often newborn lambs which cannot follow the flock. Turkish donkey owners rarely use their donkeys for ploughing, planting, cultivating, or cart traction. Donkey owners are often smallholders and poor people who cannot care for or feed their donkeys. Therefore, Turkish donkeys often are small in size and have bad body conditions [10]. Some morphometric studies have been performed in Turkish donkey populations in East and Southeast Turkey. In addition, it has been expressed among the public that there exist some Anatolian donkey breeds [10,11,12].

In one study, live weight and various body measurements were collected from 500 different donkeys from 16 different provinces of Turkey in order to determine their morphological characteristics considering province, sex, colour, and age group. The least squares mean for the animals aged 1–3 years, 4–5 years, 6–8 years, and 9 years and over for body length were obtained as 99.16 cm, 107.28 cm, 110.71 cm, and 111.02 cm, respectively. In this study, the males were determined to have higher live weight, withers height, rump height, and chest depth values than females. Significant differences in the live weight of the donkeys were related to province, age, colour, and sex [13]. In a study on the morphological characteristics of Amiata donkeys raised in Tuscany, for 6-year-old stallions, the average sizes were as follows: height at withers, 129.8 cm; thorax circumference, 145.6 cm; and front shank circumference, 18.3 cm [14].

Through the use of body measurements, morphometry assesses an animal’s forms, supporting the assessment of body conformation and the development of precise metrics to gauge an animal’s productivity [15]. According to the primary objective of the animal working (traction and/or load) or riding—some research has revealed that body measurements that determine balance and body proportionality can be defined differently [15].

Mean values of all body measurements, including back height, body length, body weight, carpal circumference, carpal height, chest circumference, chest depth, chest width, and croup length, were higher in Banat donkeys than in hybrids and individuals from both subpopulations of the Balkan donkeys. For each of the four assembled donkey groups (BanD, HY, BalkD-BGP, and BalkD-RGP), the mean body length was 131.1, 119.8, 112, and 114 cm, respectively [16].

Statistical studies have been conducted to predict morphological variables from body measurements of donkeys of various breeds and in different regions. Quadratic, Probit, Logistic, and Exponential models have been used to estimate body weight from heart girth in Miranda donkeys. These models were compared, and the best fit was obtained by the Quadratic model [17]. In another study [18], a Bayes Linear Regression Model was applied for the estimation of sperm quality parameters using age, body weight, testicular morphometry, and combined biometric indices in donkeys. Statistics were calculated from testicular ultrasonographic measurements in donkeys, precision calliper measurements after orchiectomy in some of the donkeys, and sperm quality parameters.

To the best of our knowledge, there is no research on the use of CHAID, Automatic Linear Model, Random Forest, MARS, and Bagging MARS algorithms for body length prediction using different biometric measurements from donkeys. This study aims to determine the body length of donkeys through a comparative analysis of data mining methods, utilising gender, age, coat colour, and morphological characteristics. The results are expected to be useful in determining the relationships between various morphological features of donkeys.

## 2. Materials and Methods

For this study, 371 donkeys aged 1–15 years old were analysed, with data collected from Agri (39°43′ N; 43°03′ E), Ardahan (41°06′ N; 42°42′ E), Artvin (42°14′ N; 41°49′ E), Balıkesir (39°39′ N; 27°52′ E), Erzurum (39°54′ N; 41°16′ E), Hakkari (37°34′ N; 43°44′ E), Iğdır (39°55′ N; 44°02′ E), Kars (40°36′ N; 43°06′ E), Konya (37°52′ N; 32°31′ E), Mardin (37°19′ N; 40°44′ E), Sanliurfa (37°09′ N; 38°49′ E), Sirnak (37°31′ N; 42°27′ E), and Van (38°30′ N; 43°22′ E) provinces in Turkey [19,20]. The section relates to physical description results mainly from studies carried out by the second author between October 2013 and February 2014 in 13 provinces in Turkey.

The donkeys were distributed by colour as follows: 70 brown donkeys (18.87%), 146 grey donkeys (39.35%), 81 black donkeys (21.83%), and 74 white donkeys (19.95%). In terms of gender, there were 119 female donkeys (32%) and 252 male donkeys (68%).

Live weights and various body measurements were collected from the native donkey populations in different provinces of Turkey in order to determine their morphological features.

A total of 12 different body measurements were collected from the donkeys. The body measurements, including withers height (WH), height at rump (HR), chest depth (CD), chest width (CW), and body length (BL), were measured by means of a measuring stick. Other body measurements, including chest circumference (CC), head length (HL), front shank circumference (CAC), head length (HL), and ear length (EL), were obtained by means of a tape measure [21,22]. The ages of the donkeys were determined by their owners.

### 2.1. Chi-Square Automatic Interaction Detection (CHAID)

CHAID is a technique based on a criterion variable with two or more categories. This allows investigators to determine the segmentation with respect to that variable in accordance with the combination of a range of independent variables [23,24]. CHAID was originally proposed in [25]. The CHAID algorithm applies the F significance test to a scale response variable [26], and Bonferroni adjustment is utilised to calculate adjusted *p* values in the tree structure [27].

The selection of the suitable independent variables from the set of input variables is made in such a way that, in the resulting hierarchically arranged structure, the first independent variable for the partition of input data is selected as the variable with the lowest *p*-value, and is, for this reason, most strongly associated with the dependent variable. In hypothesis testing, if the *p* value is equal to or lower than the predefined level of significance α, then the alternative hypothesis, which suggests a dependency between variables, is accepted, which, in the context of tree development, denotes node splitting using a given independent variable. Else, the node is considered to be the terminal node. Tree building ends when the *p* values of all the observed independent variables are higher than a certain split threshold [28].

Merging the values of each independent variable so that a certain number of nodes, with statistically significant differences between them, appear on the tree. Actually, the algorithm identifies pairs of values of independent variables which are least different from the dependent variable so that the number of categories of predictor variables depends on the Chi-square test results and *p* value. If the obtained *p* value is higher than a certain merge threshold, the algorithm merges particular categories with no statistically significant differences. Next, the search for a new merging pair continues till the pairs for which the *p* value is smaller than the described level of significance α, are not identified [28].

Two key functions of statistical tests in CHAID analysis can be identified. These are a combination of individual values and determination of predictor variable categories and the selection of predictor variables according to the statistical significance of their association with the dependent variable [29]. If nonbinary predictor variables are concerned, then the test value increases along with the number of branches into which they are split. However, variables with more categories are more likely to be identified as statistically significant in relation to the dependent variable compared to the independent variables with fewer categories [24].

In the ordinal classification problems for variables, the Chi-square test is used in determining the significance of the relationship and the best split for each tree building level. For regression-type problems, the F-test is used as the criterion of numerical variables division [30]. Such applicability to both classification and regression problems is one of the key advantages of this algorithm. Conversely, one of the key disadvantages of the CHAID method is that it requires large amounts of data because they are at every tree level split into several groups, which may become too small for reliable analysis [31].

### 2.2. Random Forest (RF) Algorithm

The random forest model (RF) is a premium method for regression and classification in the field of decision tree learning. It is very influential, as its regression accuracy is typically better than that of other regression methods. The RF model was proposed by L. Breiman in 1984 [32].

Instead of splitting each node using the best split among all assessed variables, RF splits each node using the best among a subset of predictors randomly chosen at that node. A new training data set is obtained from the original data set with a replacement. Then, a tree is grown using random attribute selection [33].

RF is very fast and robust against overfitting, and it is possible to form as many trees as the user wants [34]. After developing a number of decision trees, the output of the model is obtained by averaging the output values of all of the individual trees. After training single trees, the learner bagging algorithm is applied to the RF model. Bagging repeatedly selects bootstrap samples from the training set and fits t_b_ trees considering the Gini impurity of these samples. After the training process, the predicted values for unseen instances *x* are calculated by averaging the prediction results from all regression trees, as follows:$$f\left(x\right)=\frac{1}{B}\sum_{b=1}^{B}t_{b}\left(x\right).$$

Here, a random forest (RF) was learnt from an ensemble of 500 regression trees. All variables were included as predictors and total tree height served as the response variable [35]. The RF method was built using the R function “ranger”, with all hyperparameters kept at the default values. The “Ranger” function was applied as a faster and more memory-influential random forest implementation for the analysis of data, compared to other commonly used random forest packages in R [36].

To model the relationship between body morphological features in this study, given these to training input–output, the RF regression model was performed as follows:

From the initial training dataset, ntree bootstrap sample sets, or $X_{i}$ (*i* = bootstrap iteration, and its value was restricted to the range of [1, ntree]), were drawn at random with replacement. For that bootstrap sample set, the elements that are absent from *X_i_* are referred to as out-of-bag data.Morphological features were randomly chosen at each node of each tree, and the feature with the lowest Gini Index that best partitioned those features was chosen.For each tree, until a predetermined stop condition was met, the data splitting process in each internal node of a rule was repeated from the root node.

### 2.3. Multivariate Adaptive Regression Spline (MARS)

Multivariate adaptive regression spline (MARS) is a nonparametric regression method put forward by Friedman in 1991 [37,38]. A regression double is typically denoted by ($X_{i}$, $Y_{i}$), where $X_{i}$ represents the independent variable(s), and $Y_{i}$ represents the dependent variable(s). In the MARS model, there are one or more split point(s) for every independent variable, denoted as $t_{i}$. For $X_{i}$ ≥ $t_{i}$, there is an equation named the right-side basis function (BF), while for $X_{i}$ < $t_{i}$ there is another equation named the left-side basis function. These two basis functions (spline functions) relate $X_{i}$ to the dependent variable $Y_{i}$. The following equations provide the mathematical representation of the right and left basis functions [39]:$${\left[-\left(X_{i}-t_{i}\right)\right]}_{+}^{q}=\left\{\begin{array}{c} {\left(t_{i}-X_{i}\right)}^{q}\;If\;X_{i}<t_{i}\\ 0\;otherwise \end{array}\right.,$$
$${\left[+\left(X_{i}-t_{i}\right)\right]}_{+}^{q}=\left\{\begin{array}{c} {\left(X_{i}-t_{i}\right)}^{q}\;If\;X_{i}\geq t_{i}\\ 0\;otherwise \end{array}\right.,$$
where q (≥0) is the power to which the splines are raised, which defines the degree of smoothness of the outcome function estimate.

In a MARS algorithm, the approximated MARS function is composed of a linear combination of basis functions, defined as a product of basis functions. The MARS model can be written as follows [40]:$$f\left(x\right)=\sum_{m=1}^{m}a_{m}B_{m}\left(x\right),$$
where *f*(*x*) is the MARS model and $B_{m}\left(x\right)$ is called the basis function. Here, *m* denotes the index of the basis function, while m indicates the total number of basis functions in the MARS model. The coefficient of the *m*th basis function is written as *a_m_*, and $x\;\epsilon \;R_{n}$ denotes the predictor variable vector. MARS utilises a product form for the basis function:$$B_{m}\left(x\right)=\prod_{k=1}^{k_{m}}b_{km},$$
where $b_{km}$ is the *k*th univariate function in $B_{m}\left(x\right)$ and $k_{m}$ denotes the total number of univariate terms multiplied in $B_{m}\left(x\right)$. When $k_{m}>1$, then $k_{m}$ is the degree of the interaction term. Conversely, if $k_{m}=1$, then the basis function is univariate. In each basis function, the refraction points are the knots for the basis function. The simplest form for $b_{km}$ are truncated linear functions of the form:$$b(x/t)={\left[+\left(x-t\right)\right]}_{+}=max\left\{+\left(x-t\right),0\right\}$$
or
$$b(x/t)={\left[-\left(x-t\right)\right]}_{+}=max\left\{-\left(x-t\right),0\right\},$$
where the location *t* is called the knot of the basis function.

In order to eliminate duplicate BFs, MARS uses the generalized cross-validation (*GCV*) criteria, which is stated in the following way [41]:$$GCV=\frac{\frac{1}{N}\sum_{i=1}^{n}{\left[y_{i}-f^{\prime }(x_{i})\right]}^{2}}{{\left[1-\frac{C(B)}{N}\right]}^{2}}.$$

The MARS predictive model with interaction term used in this paper was constructed based on the lowest GCV [42]. Ten-fold cross validation was considered as a resampling technique in the MARS model.

Here, *N* represents the total number of points in the data, while *C*(*B*) is a complexity penalty that increases with the number of BFs in the model, determined as follows [43]:
*C*(*B*) = (*B* + 1) + *d*(*B*).

### 2.4. Bootstrap Aggregating Multivariate Adaptive Regression Splines (Bagging MARS)

The Bagging MARS algorithm is a technique put forward in [44], which is performed to reduce the variance estimators in classification and regression. The use of this method is not only limited to improving the estimator but it may also be used to improve the accuracy and predictive power. Bagging MARS algorithm usages bootstrapping among resampling methods. Bagging models may provide their own internal estimate of predictive accuracy correlating well with either cross-validation estimates or test set estimates [45].

Theoretically, the bagging estimator is described as $f_{Bagging}=E\left[\hat{f}(x)\right]$ [44]. In practice, the bootstrap expectation is obtained through the use of a Monte Carlo method. For every bootstrap simulation $b\epsilon \left\{1,2,\ldots ,B\right\}$, the MARS method ${\hat{f}}_{b}(x)$ is calculated to approximate the bagging expectation as follows:$${\hat{f}}_{Bagging}=\frac{1}{B}\sum_{b=1}^{B}{\hat{f}}_{b}(x),$$
where the number *B* indicates the accuracy of the Monte Carlo approximation. Its value is generally taken as 100, depending on the sample size.

A bootstrap sample (*n*) is a sample acquired randomly from the studied data on the basis of replacement. Some data points are selected multiple times in the bootstrap sample. Bagging MARS is a useful tool that is used to enhance the predictive accuracy of the MARS model [46]. Here, number of bootstrap samples was considered as three.

### 2.5. Automatic Linear Model (ALM)

Automatic Linear Modelling (ALM) is an improved version of the linear regression method, which is used to process and analyse data and make predictions. The term ALM refers to a data mining approach similar to regression trees, which utilises a machine learning approach to determine the best predictive model using the existent data. ALM is carried out in several steps, including preliminary data processing, replacing missing data values, determining the quality predictor, identifying outliers, and calculating the stepwise model and coefficient of determination (*R*^2^).
$$f\left(x\right)=y=c+b_{1}x_{1}+b_{2}x_{2}+\cdots +b_{n}x_{n},$$
where *y* is the dependent variable; *c* is a constant; *b*_1_, *b*_2_, …, *b_n_* are the parameter coefficients; and *x*_1_, *x*_2_, …, *x_n_* are the independent variables [47].

Since the process of evaluating all possible subsets can provide the best subsets after taking into account all possible regression models, the researcher can then select an appropriate final model from the most promising subsets [48].

ALM is considered a new method, introduced in SPSS software (version 19 and higher), and allows researchers to automatically select the best subset when there are generally large numbers of variables. In ALM, prediction variables are automatically transformed to provide an improved data fit, and SPSS uses time and other metrics rescaling, outlier correction, and other methods for this purpose [49].

To compare the predictive performances of the CHAID, RF, MARS, and Bagging MARS models in the 10-fold cross-validation, the following model evaluation criteria were calculated [50,51,52,53]. Performance evaluation of used data mining techniques is performed with proportions 80:20 of training data and test data.

Coefficient of determination:$$R^{2}=1-\frac{\sum_{i=1}^{n}{\left(Y_{i}-{\hat{Y}}_{i}\right)}^{2}}{\sum_{i=1}^{n}{\left(Y_{i}-\bar{Y}\right)}^{2}}.$$

Root-mean-square error (*RMSE*):$$RMSE=\sqrt{\frac{1}{n}\sum_{i=1}^{n}{\left(Y_{i}-{\hat{Y}}_{i}\right)}^{2}}.$$

Mean absolute percentage error (*MAPE*):$$MAPE=\frac{1}{n}\sum_{i=1}^{n}\left|\frac{Y_{i}-{\hat{Y}}_{i}}{Y_{i}}\right|\times 100.$$

Standard deviation ratio:$${SD}_{ratio}=\sqrt{\frac{\frac{1}{n-1}\sum_{i=1}^{n}{\left(\varepsilon_{i}-\bar{\varepsilon }\right)}^{2}}{\frac{1}{n-1}\sum_{i=1}^{n}{\left(Y_{i}-\bar{Y}\right)}^{2}}}.$$

The R software version 4.2.0 was used for the analyses, taking the number of folds in the cross-validation as 10 [54]. Results were obtained using the RF algorithm “randomForest”, MARS and Bagging MARS algorithms in the “earth” packages, while the model evaluation performance criteria for the data mining algorithms were calculated using the “ehaGoF” package [55].

Using the R “corrplot” package, the Pearson correlation coefficients between BL and body characteristics were calculated. Furthermore, the multicollinearity problem between the independent variables was assessed at the outset of the analysis, and it was discovered that there was no issue. CHAID and ALM techniques were carried out using relevant packages in the SPSS V.26.0 software (2019) [56].

## 3. Results

### 3.1. Descriptive Statistics

Descriptive statistics, including the morphological characteristics, of donkeys aged 1–15 years with 4 different hair colours and bred in 13 different cities in Turkey are given in Table 1.

### 3.2. Correlation Matrix and Principal Component Analysis (PCA) Results

The correlation matrix for the body morphological characteristics of donkeys is presented in Table 2.

Examining the correlation coefficients in Table 2, the correlation coefficients among all morphological features were found to be positive. The highest correlations were between WH and HR (0.951), WH and HL (0.783), and BL and HR (0.767); meanwhile, the weakest correlations were between EL and HL (0.045), CD and HL (0.096), and TL and EL (0.126). This information is also confirmed in the Principal Component Analysis (PCA) graph shown in Figure 1. According to the PCA analysis, the contribution of principal component 1 (PC1) was 53.11%, while that of principal component 2 (PC2) was 10.53%, for a total of 63.64%. In the PCA analysis, it was determined that all variables were in the same direction, and the correlation coefficients between all variables were positive. As a result of the PCA, it was found that CAC and LL, HW and CW, HR and WH, CD and CC, and EL and HL were closely related to each other. In other words, the correlation coefficients between closely related variables were high and important. TL and HL are very distantly related, and therefore, the correlation coefficient between them was the smallest. Similarly, CD and HL and EL and TL were also distantly related, thus presenting low correlation coefficients.

The CHAID, random forest (RF), automatic linear modelling (ALM), MARS, and Bagging MARS methods were analysed to determine the effects of other morphological features on body length in donkeys. Their respective results are summarised in the following.

### 3.3. Result of the CHAID Algorithm

In order to determine the effects of variables on body length, the parent node/child node ratio was set as 32:16 in the CHAID algorithm. The number of folds in the cross-validation was set as 10, and the regression tree obtained by the CHAID algorithm is presented as a diagram in Figure 2.

Examining the CHAID diagram (Figure 1), it can be determined that the first-order effective independent variable affecting the body length of donkeys was HR (Adj. *p* value = 0.000, F = 132.422); the second-order independent variables were TL (Adj. *p* value = 0.001, F = 17.781), CW (Adj. *p*-value = 0.000, F = 30.249), and HW (Adj. *p* value = 0.000, F = 23.959); and the third-order independent variables were CAC (Adj. *p* value = 0.000, F = 23.827), LL (Adj. *p* value = 0.003, F = 14.527), and HW (Adj. *p* value = 0.048, F = 6.925). Branches generated by independent variables in the whole tree construction were statistically significant (*p* < 0.05). The performance of the CHAID algorithm was calculated as 0.728, 0.521, 3.89, and 3.03 in terms of R^2^, standard deviation ratio (SD ratio), RMSE, and MAPE, respectively. The results of the CHAID algorithm are generally summarised as follows.

The HR variable is divided into five sub-nodes. If HR ≤ 96 cm, mean BL = 94.300 cm (Node 1). If 96 < HR ≤ 102 cm, the mean BL = 101.747 cm (Node 2). If 102 < HR ≤ 106 cm, the mean BL = 103.816 cm (Node 3). If 105 < HR ≤ 110 cm, the mean BL = 106.328 cm (Node 4). If HR > 110 cm, the mean BL = 118.081 cm (Node 5). In short, as the height at rump (HR) value increased, the body length (BL) also increased.

When HR ≤ 96 cm, TL ≤ 41 cm, mean BL = 91.864 cm, while TL > 41 cm, mean BL = 97.278 cm was obtained. When 96 < HR ≤ 102 cm, the mean BL value was estimated as 98.636 cm if CW ≤ 26 cm, 100.721 cm if 26 < CW ≤ 28, and 104.448 cm if CW > 28 cm.

When CAC ≤ 12.500 cm, BL = 95.667 and CAC > 12.500 cm, BL = 100.692 cm was estimated. When 26 < CW ≤ 28, LL ≤ 49 cm, BL = 99.240 and LL > 49 cm, BL = 102.778 cm was estimated. When CW > 28, if LL ≤ 49 cm, BL = 103.400 and LL > 49 cm, BL = 107.529 cm was estimated. When 102 < HR ≤ 106 cm and CW ≤ 29 cm, BL = 101.412. When 102 < HR ≤ 106 cm and CW > 29 cm, BL = 105.762. When CW > 29 cm, BL = 103.895 cm when HW ≤ 36 cm and BL = 107.304 cm when HW > 36 cm. If 105 < HR ≤ 110 cm and HW ≤ 35 cm, then BL = 103.111 cm. If 105 < HR ≤ 110 cm and HW > 35 cm, then BL = 108.676 cm.

### 3.4. Random Forest (RF) Algorithm Results

The RF algorithm results are summarised as follows. The random forest trees created to obtain the smallest error value are presented in Figure 3.

The model was constructed using an RF algorithm with the dependent variable as body length (BL). In the RF model, the linear traits of animals were included as predictors, namely, WH, HR, CD, CW, CC, HL, CAC, HL, EL, Province, sex, and coat colour. The random forest algorithm included 500 trees. The model described 82.95% of the variation of the dependent variable, with MSE = 8.911, RMSE = 2.985, MAE = 2.4, and Bias = 0.6. In the constructed model, the most significant factor affecting body length was province, withers height (WH), followed by HR and HL, respectively (Table 3 and Figure 4).

The regression tree depicting the morphological features affecting BL in the random forest (RF) algorithm is shown in Figure 5.

Examining Figure 5, the following explanations can be derived from the RF algorithm (n = 297 nodes).

split, n, deviance, yval, * denotes terminal node(1) root 297 16845.50000 103.76430(2) HL < 60 255 8246.78400 101.72550(4) HR < 97.5 44 1127.90900s 94.95455(8) CC < 96.5 12 160.00000 90.00000 *(9) CC >= 96.5 32 562.87500 96.81250(18) CW < 28.5 23 185.21740 95.34783 *(19) CW >= 28.5 9 202.22220 100.55560 *(5) HR >= 97.5 211 4681.01400 103.13740(10) CW < 30.5 156 2869.91700 101.91670(20) LL < 49.5 102 1540.87300 100.42160(40) CAC < 14.25 95 1260.00000 100.00000(80) CW < 28.5 60 405.65000 98.65000 *(81) CW >= 28.5 35 557.54290 102.31430(162) Province = Artvin, Kars, Sanliurfa, Sirnak 7 76.85714 97.14286 *(163) Province = Ardahan, Balikesir, Erzurum, Hakkari, Igdir, Konya, Van 28 246.67860103.60710 *(41) CAC >= 14.25 7 34.85714 106.14290 *(21) LL >= 49.5 54 670.37040 104.74070(42) LL >= 51.5 20 140.55000 102.35000 *(43) LL < 51.5 34 348.26470 106.14710(86) CC < 112 12 50.00000 103.00000 *(87) CC >= 112 22 114.59090 107.86360 *(11) CW>= 30.5 55 919.20000 106.60000(22) HW < 38.5 45 424.31110 105.35560 *(23) HW >= 38.5 10 111.60000 112.20000 *(3) HL >= 60 42 1103.14300 116.14290(6) CD < 47.5 16 180.00000 111.50000 *(7) CD >= 47.5 26 366.00000 119.00000 *

### 3.5. Automatic Linear Modelling (ALM) Results

The model prediction coefficients and significance values obtained from the ALM are provided in Table 4.

The ALM evaluated the predictability of the body length mean score. The morphological features which contributed most to the model are shown in Table 4. Notably, the variables EL and sex were not statistically significant in the ALM procedure. Table 4 also shows estimates for the parameters included in the overall model and their individual effects on the target variable. The coefficients focused on the relationship that each predictor had with the mean body length, holding the values of other predictor variables constant. The importance values of the predictors, as defined by the ALM procedure, are also given in Table 4. These values were normalised, such that the importance values were summed to 1. The accuracy value of this model was 71.3% (i.e., the adjusted R^2^ of the model multiplied by 100).

The predictor importance graph (Figure 6) indicates the relative importance of each predictor in estimating the model, where the values for the province, HR, CC, HL, HW, TL, and CAC were 0.415, 0.231, 0.171, 0.065, 0.051, 0.034 and 0.033, respectively. Overall, the results indicate that province was the most important predictor of body length.

The discarded scatter plot for BL (Figure 7) displays predictor values on the *y*-axis and observed values on the *x*-axis, and it can be seen that a higher percentage of the sample locations lie on the 45-degree line; therefore, the model was reasonably accurate. Cook’s distance of body length (PL) identified that sample locations such as 323 (0.043), 219 (0.041), 28 (0.034), 209 (0.032), 45 (0.028), 50 (0.022), 55 (0.021), 292 (0.020), 282 (0.016), 303 (0.016), 54 (0.015), 3 (0.012) and 240 (0.012) had the highest values (Table 5). The predictor values further indicated that the province, HR, CC, HL, HW, TL, and CAC were positively correlated with BL (Figure 8).

### 3.6. MARS Algorithm Results

The model estimation coefficients obtained by the MARS algorithm for the prediction of body length are given in Table 6.

According to the results presented in Table 6, all of the coefficients for the MARS predictive model were statistically significant (*p* < 0.001). The desirable predictive quality of the MARS equation produced here was obtained while ensuring the smallest GCV (7.862). The recorded or observed values in body length of donkeys were correlated very strongly with those predicted by the MARS model (*p* < 0.001) as an animal breeding model. For the prediction equation of the MARS model with 50 terms, no over-fitting was observed, as the R^2^ estimate (0.916) was close to the CVR^2^ estimate (0.859). The SD ratio of 0.291, RMSE of 2.173, and MAPE of 1.615 indicate that the MARS model for capturing influential factors, such as morphological characteristics and age, had an excellent fit.

From the MARS algorithm results, some terms and their coefficients can be interpreted as follows: When HR > 108 in donkeys, the effect (corresponding to the positive coefficient of 0.286) on body length was found to be positive; if HR ≤ 108, the corresponding negative coefficient (−1.063314) on body length was found to result in an adverse effect. If CC > 115, the effect on body length (BL) was positive (4.494783). If CC > 116, the effect on body length (BL) was negative (−8.182082). If CC ≤ 118, the effect on body length (BL) was positive (0.977991). If CC > 118 cm, BL increases by 13.999379 cm. If CC ≤ 119 cm, the BL value decreases by 7.825563 cm. When HW ≤ 41 cm, BL decreases by 1.210453. When HW > 41 cm, the BL value decreases by 1.296884 cm. If TL ≤ 51 cm, BL decreases by 0.138144 cm. For donkeys bred in Mardin province, if WH ≤ 106 cm, BL increases by 0.443792 cm. If HR ≤ 98 cm and CC ≤ 118 cm, BL decreases by 0.065563. When HR ≤ 108 cm and HW ≤ 36 cm, BL increases by 0.144160 cm. When HR ≤ 108 cm and CAC > 13.5 cm, BL increases by 0.906941 cm. When HR ≤ 108 cm and LL > 45 cm, BL increases by 0.055649 cm. When CC > 113 cm and HW ≤ 41 cm, the BL value increases by 0.908293. When CC ≤ 114 cm and HW ≤ 41 cm, the BL value decreases by 0.111693. When CC > 114 cm and HW ≤ 41 cm, the BL value decreases to 1.079222. When CC > 118 cm and TL > 42 cm, the BL value increases by 0.116470. If CC > 118 cm and TL ≤ 42 cm, the BL value increases by 0.304491. When HW ≤ 41 cm and HL > 59 cm, BL increases by 0.211118 cm. If HW ≤ 41 cm and HL ≤ 59 cm, BL increases by 0.061253 cm. If HW ≤ 41 cm and HL > 59 cm of donkeys bred in Kars province, BL decreases by 0.052772 cm. If Age ≤ 2, HW ≤ 41 cm and HL ≤ 59 cm, BL decreases by 0.049791 cm. If HR ≤ 108 cm, CW ≤ 25 and CAC ≤ 13.5 cm, BL increases by 0.067626 cm. If HR ≤ 108 cm, CW ≤ 25 and CAC ≤ 13.5 cm, BL increases by 0.088635 cm. If HR ≤ 108 cm, HW > 36 cm and CAC > 13.5 cm, BL increases by 0.534718 cm. If HR ≤ 108 cm, HW > 36 cm and TL ≤ 13 cm, BL increases by 0.556636 cm. If HR ≤ 108 cm, TL ≤ 46 cm and CAC ≤ 13.5 cm, BL increases 7.388817 cm. If CC > 113 cm, HW ≤ 41 cm and LL ≤ 46 cm, BL decreases by 0.312434 cm. If CC > 113 cm, HW ≤ 41 and LL ≤ 46 cm, BL decreases by 0.312434 cm. If HR ≤ 108 cm, TL ≤ 46 cm and CAC > 13.5 cm, BL decreases by 0.066624 cm. If HR ≤ 108 cm, CC > 111 cm, HW > 36 cm, and LL > 48 cm, BL decreases by 0.016711 cm.

The greatest positive effect on body length in donkeys was 13.999 cm when CC > 118 cm; the second-largest positive effect was when HR ≤ 108, TL ≤ 46, and CAC > 13.5, in which case the body length will increase by 7.389 cm. The third-largest positive effect was when CC > 115 cm, where body length will increase by 4.495 cm. As for the greatest negative effect, body length will decrease by 8.18 cm when CC >116 cm. The second- and third-largest negative effects on body length were −7.826 cm if CC >116 cm and −1.297 cm when HW > 41 cm, respectively.

The equation obtained by including the interaction effects of the coefficients in the model is given in detail below.
BL= 105.9362 − 1.063314 * max(0, 108 − HR) + 0.2863338 * max(0, HR − 108)  + 4.494783 * max(0, CC − 115) − 8.182082 * max(0, CC − 116)  + 0.9779915 * max(0, 118 − CC) + 13.99938 * max(0, CC − 118)  − 7.825563 * max(0, CC − 119) − 1.210453 * max(0, 41 − HW)  − 1.296884 * max(0, HW − 41) − 0.138144 * max(0, 51 − TL)  + 0.4437916 * Province Mardin * max(0, WH − 106)  − 0.06556272 * max(0, 98 − HR) * max(0, 118 − CC)  + 0.1441603 * max(0, 108 − HR) * max(0, 36 − HW)  + 0.906941 * max(0, 108 − HR) * max(0, CAC − 13.5)  + 0.05564873 * max(0, 108 − HR) * max(0, LL − 45)  + 0.04835515 * max(0, 108 − HR) * max(0, 45 - LL)  + 0.9082933 * max(0, CC − 113) * max(0, 41 − HW)  − 0.1116934 * max(0, 114 − CC) * max(0, 41 − HW)  − 1.079222 * max(0, CC − 114) * max(0, 41 − HW)  − 0.1164698 * max(0, CC − 118) * max(0, TL − 42)  + 0.3044909 * max(0, CC − 118) * max(0, 42 − TL)  + 0.2111183 * max(0, 41 − HW) * max(0, HL − 59)  + 0.06125319 * max(0, 41 − HW) * max(0, 59 − HL)  − 0.05277187 * Province Kars * max(0, 41 − HW) * max(0, 59 − HL)  − 0.04979143 * max(0, 2 − Age) * max(0, 41 − HW) * max(0, 59 − HL)  + 0.06762594 * max(0, 108 − HR) * max(0, CW − 25) * max(0, 13.5 − CAC)  + 0.08863474 * max(0, 108 − HR) * max(0, 25 − CW) * max(0, 13.5 − CAC)  + 0.5347178 * max(0, 108 − HR) * max(0, HW − 36) * max(0, CAC − 13)  + 0.5566363 * max(0, 108 − HR) * max(0, HW − 36) * max(0, 13 − CAC)  + 7.388817 * max(0, 108 − HR) * max(0, 46 − TL) * max(0, CAC − 13.5)  − 0.3124338 * max(0, CC − 113) * max(0, 41 − HW) * max(0, 46 − LL)  − 0.06662377 * max(0, 108 − HR) * CC * max(0, 46 − TL) * max(0, CAC − 13.5)  − 0.01671139 * max(0, 108 − HR) * max(0, CC − 111) * max(0, HW36) * max(0, LL − 48) 

It is also possible to estimate the body length by assigning various values to the morphological features that express the independent variables in the equation obtained using the MARS algorithm. For example, with Age = 12, WH = 107, HR = 106, CC = 124, CD = 47.5, CW = 34, HW = 41.5, TL = 59.5, HL = 55.5, CAC = 14.5, LL = 51, and EL = 22.5, when Colour = “Grey“, Province = ”Konya”, and Sex = “Male”, we obtain BL = 114.932 cm.

The relative importance of the variables predicting body length as a result of the MARS algorithm is demonstrated in Figure 9. According to the MARS algorithm, the predictor importance graph displays the relative importance of each predictor in estimating the model, where the values for HR, WH, HL, province, HW, CW, LL, CC, CD, CAC, TL, EL, and sex, respectively. Overall, the results indicate that the HR variable was the most important predictor of body length.

The estimated values obtained by the MARS algorithm are presented together with the observed values in Figure 10.

### 3.7. Bagging MARS Algorithm Results

The prediction equation and detailed results of the Bagging MARS algorithm are included as Appendix A.

According to the Bagging MARS algorithm results, in the first bootstrap, an increase in body length can be expected for those with WH > 105 cm, CC > 107 cm, HR > 99 cm with CW ≤ 29 cm, HR ≤ 104 cm with HW ≤ 41 cm, HR > 99 cm with TL ≤ 39 cm, HW ≤ 41 cm with HL > 58 cm, HW ≤ 41 cm with CAC > 13.5 cm, HW ≤ 41 cm with LL > 45 cm, HW ≤ 41 cm with EL > 22.5 cm, and TL ≤ 50 cm with EL ≤ 21 cm. In the second bootstrap, an increase in body length can be expected for those with HR > 107 cm, TL ≤ 38 cm, WH ≤ 101 cm with HR ≤ 107 cm, WH > 101 cm with HR ≤ 107 cm, HR ≤ 107 cm with CW > 26 cm, HR ≤ 107 cm with CW ≤ 26 cm, HR ≤ 107 cm with CAC > 13.5 cm, HR ≤ 107 cm with EL ≤ 24 cm, and HW ≤ 41 cm with HL > 59 cm. In the third bootstrap, WH > 104 cm, CC > 118 cm, CD ≤ 46 cm, TL > 41 cm, Province = “Mardin” and HW ≤ 40 cm, Age > 2 and HW ≤ 40 cm, Age > 2 and TL ≤ 41 cm, WH ≤ 104 cm and CAC > 13.5 cm, WH ≤ 104 cm and CAC ≤ 13.5 cm, CC ≤ 112 cm and CD > 46 cm, CC ≤ 118 cm and EL > 24 cm, HW ≤ 40 cm and TL ≤ 59 cm, HW ≤ 40 cm and HL ≤ 57 cm, HW ≤ 40 cm and LL ≤ 48 cm, and HW ≤ 40 cm and LL ≤ 48 cm were predicted to lead to an increase in body length.

In the first bootstrap, there was the greatest positive contribution to body length when WH was >105 cm (0.7055595). The greatest negative effect was in the case of HR ≤ 99 cm (−2.17283). In the second bootstrap, the greatest positive effect on body length was when HR > 107 cm (2.483933), while the greatest negative effect was when HW ≤ 41 cm (−1.116736). In the third bootstrap, the greatest positive effect on body length was when CC > 118 cm (8.844629), while the greatest negative effect was when CC > 119 cm (−4.048582).

The performance indicators for the Bagging MARS algorithm were calculated as 0.831, 0.384, 2. 868, and 2.122 for R^2^, SD ratio, RMSE, and MAPE, respectively.

## 4. Discussion

In a previous study [16], the mean body length (BL) was 131.4 cm (59–184 cm) in the Miranda donkey breed. According to the descriptive statistics of donkeys raised in Iğdır, a withers height of 99.1 cm, height at rump of 101.0 cm, body length of 103.0 cm, chest depth of 45.4 cm, chest width of 29.1 cm, limb length of 53.7 cm, front shank circumference of 13.4 cm, head length of 48.4 cm, and ear length of 21.8 cm were obtained [57]. These body morphological features are consistent with those observed in this study. In another study, the average weight of West African donkeys was 126 kg, with an average height at the withers of 99.5 cm and a body length of 104.4 cm [58]. This was also very close to the results obtained in this study.

The mean body length of Turkish native breed male and female donkeys with different coat colours was 101–109 cm [14], higher than that observed in this study. In one study [13], the average height at withers calculated in adult female Amiata donkeys reared in Tuscany was 125.8 cm, and their front shank length was 16.9 cm, again higher than the values obtained in this study. The mean chest circumference was found to be 108.42 cm in 6-month-old Pêga donkeys [59]. In this study, in which different body colours were studied, we obtained similar values.

In the study [14], one-way ANOVA and Tukey’s tests were used to assess the statistical significance of differences between morphological characteristics of the studied donkey groups—Banat donkey, hybrid individuals, and two sub-populations of Balkan donkeys delineated based on their nuclear genetic profiles (BalkD-BGP and BalkD-RGP). Tukey’s test in the one-way ANOVA analysis indicated that statistically significant differences between the two subpopulations of the Balkan donkey were obtained for characteristics such as body length, chest circumference, chest depth, chest width, and height at withers. The body measurements obtained from Banat donkeys were different from the results presented in the current study, while those in hybrid individuals, BalkD-BGP, and BalkD-RGP donkeys were similar.

For adult donkeys, withers height was 131.1 cm [60]. The minimum withers height for females as determined by [61] was 120 cm. In the study of [62], the mean withers height for male Pêga donkeys was reported as 131 cm. [63,64] investigated the Nordestino donkey breed and reported the withers height values as 117 cm and 106 cm, respectively, and reported that Pêga donkeys were taller. It was found to be higher than the values in this study. These differences were caused by different breeds, environmental conditions, and breeding in different regions.

When the morphological features of donkeys were examined in the study of [65], the average body length was 64 cm, chest circumference 113.2 cm, height at withers 102.4, tail length 60.7 cm, and ear length 26.7 cm. When compared with the body measurements in this study, chest circumference and height at withers measurements were found to be very close to each other, tail length and ear length characteristics were higher, but body length values were lower. While the correlations between these variables in the authors’ study were in the range of −0.50–0.85, in this study, the correlation coefficients between the same variables were obtained in the range of 0.126–0.951. In different continents or regions where animals were raised, climate differences such as temperature and precipitation played an important role in this difference. In addition, the difference in donkey populations in the studies can be considered as another factor in the different results.

## 5. Conclusions

In this study, the performance indicators of the CHAID, RF, MARS, Bagging MARS, and ALM methods were analysed, in terms of their donkey body length prediction ability. A total of 11 morphological variables, as well as province, age, sex, and coat colour, were taken as inputs to build the models. The results were compared through several comparative statistics, including coefficient of determination (R^2^), root-mean-squared error (RMSE), mean absolute percentage error (MAPE), and standard deviation ratio (SD ratio). The outcomes of this study are as follows:

According to the MARS algorithm results, fourteen predictor variables affect body length in donkeys: namely, height at withers, height at the rump, chest circumference, chest depth, chest width, haunch width, ear length, head length, front shank circumference, limb length, tail length, age, sex, and coat colour. The variables that presented the largest contributions were chest circumference, height at rump, ear length, and front shank circumference.

The number of bootstrap samples was taken as three in Bagging MARS, which is a useful tool that can be used to improve the predictive accuracy of MARS models. However, the MARS algorithm obtained better donkey body length prediction results.

The RF method was effective in predicting the body length of donkeys, capturing 82.95% of the variation. Meanwhile, the accuracy value of ALM was 73.1%, lower than that of the RF model.

In order of importance, the variables affecting the body length in donkeys were Province, WH, and HR for the RF Algorithm; HR, WH, and HL for the MARS Algorithm; and, HR followed by TL, CW, and HW for the CHAID algorithm.

In terms of the performance results, the algorithms followed the order MARS > Bagging MARS > Random Forest > CHAID > ALM (best to worst).

Through the use of livestock data, it was concluded that data mining methods are very useful for determination of the relationships between body morphological properties, potentially allowing for the estimation of any variable.

## Figures and Tables

**Figure 1 animals-13-02366-f001:**
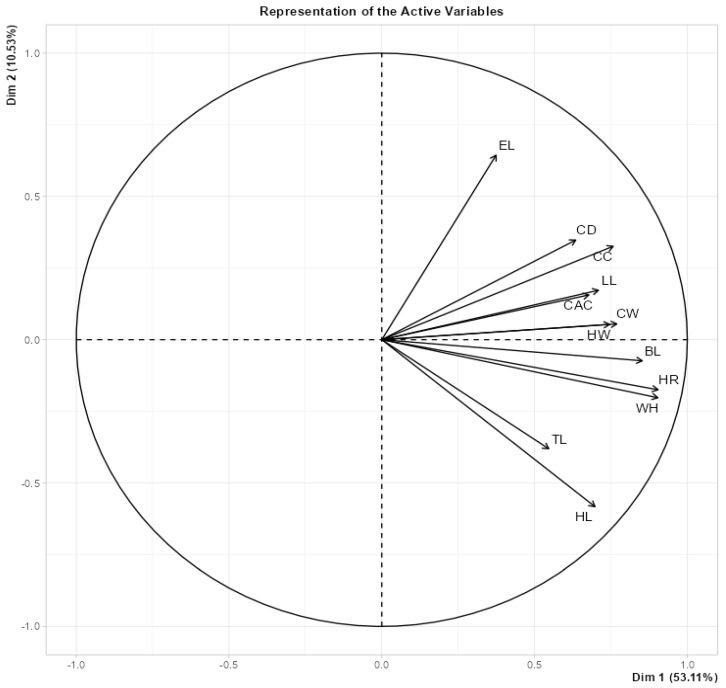
Relationships among genotypes and traits, according to the principal component analysis results.

**Figure 2 animals-13-02366-f002:**
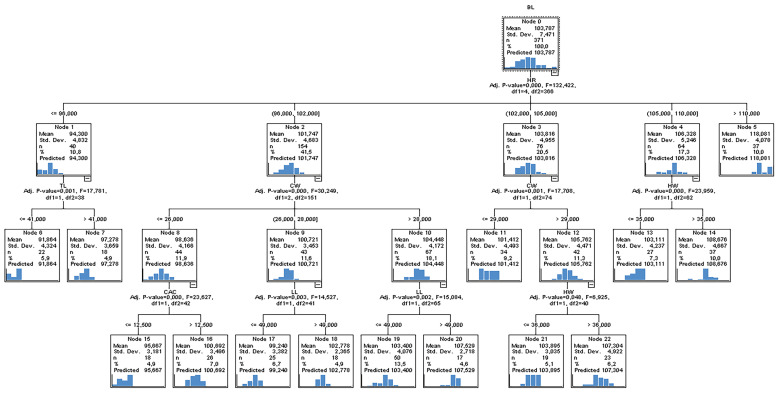
CHAID classification tree diagram for body length estimation.

**Figure 3 animals-13-02366-f003:**
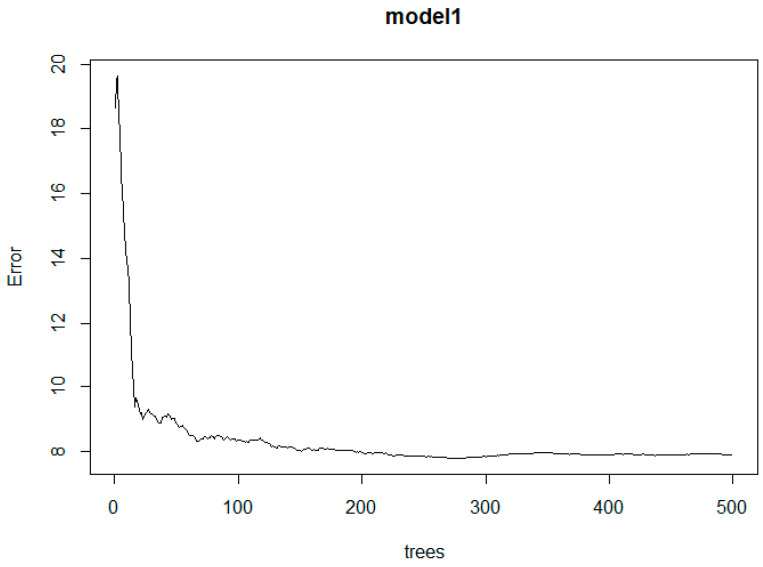
RF Algorithm error rate of the model.

**Figure 4 animals-13-02366-f004:**
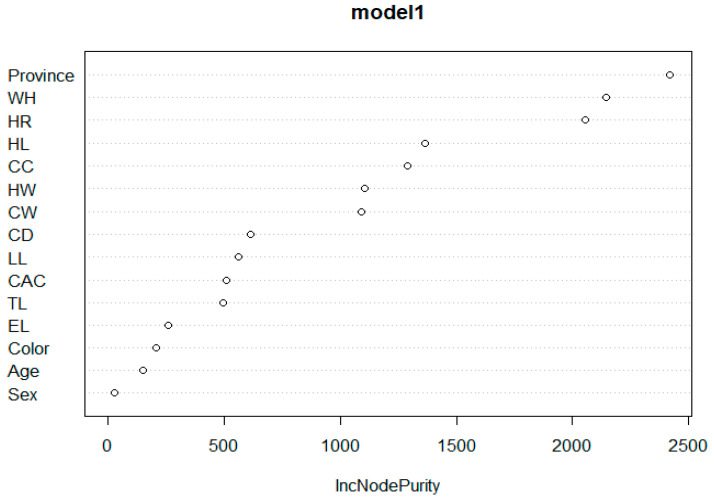
Significance graph for the variables used in the RF model.

**Figure 5 animals-13-02366-f005:**
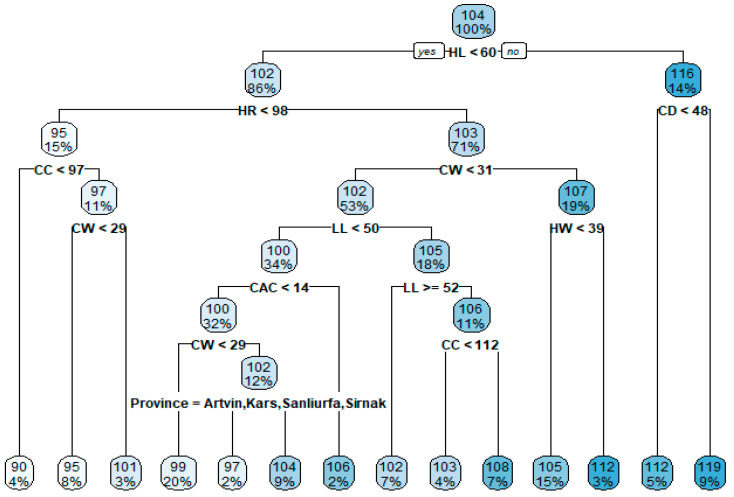
RF algorithm regression tree.

**Figure 6 animals-13-02366-f006:**
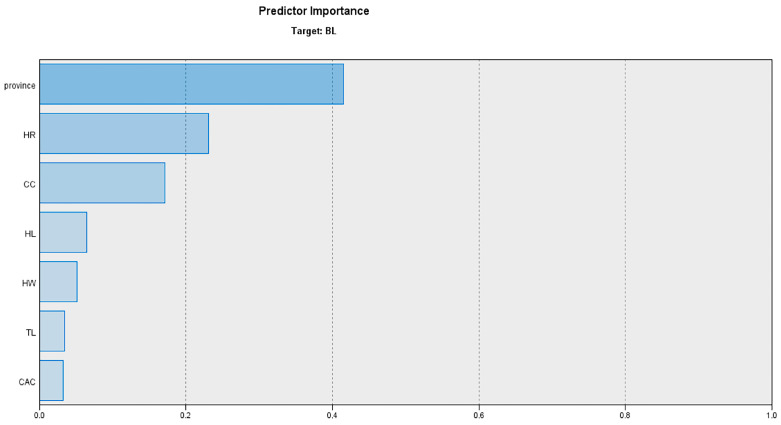
Predictor importance in the body length estimation model.

**Figure 7 animals-13-02366-f007:**
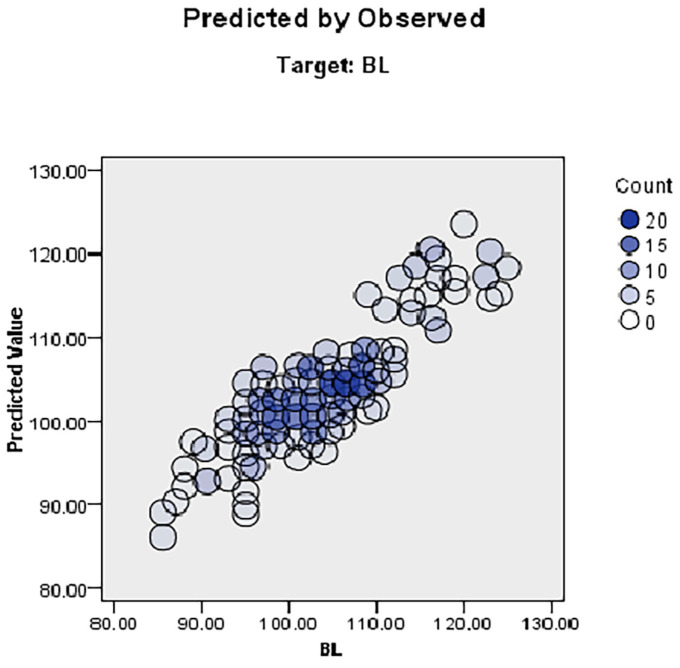
Discarded scatterplot of observed and predicted values for BL.

**Figure 8 animals-13-02366-f008:**
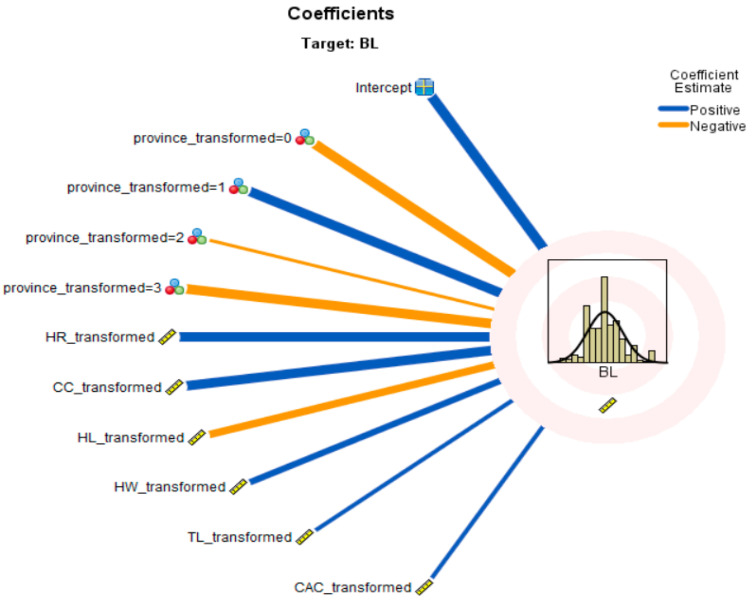
Coefficient values for BL.

**Figure 9 animals-13-02366-f009:**
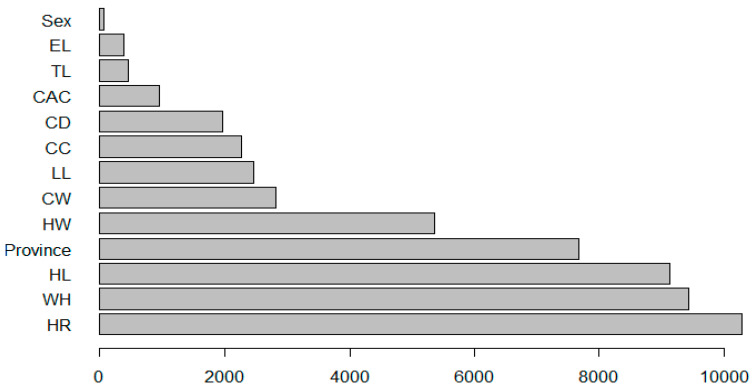
Relative importance of variables predicting body length in the MARS algorithm.

**Figure 10 animals-13-02366-f010:**
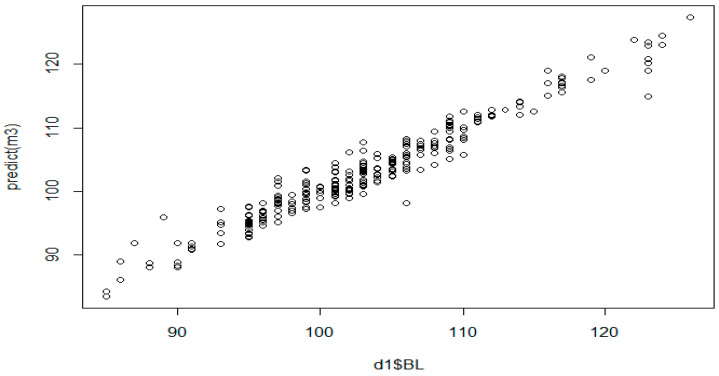
The concurrence between MARS-predicted and actual BW values.

**Table 1 animals-13-02366-t001:** Descriptive statistics regarding body morphological characteristics of donkeys.

	Colour	Sex	BL	WH	HR	CC	CD	CW	HW	TL	HL	CAC	LL	EL
N	Brown	Male	44	44	44	44	44	44	44	44	44	44	44	44
		Female	26	26	26	26	26	26	26	26	26	26	26	26
	Grey	Male	91	91	91	91	91	91	91	91	91	91	91	91
		Female	55	55	55	55	55	55	55	55	55	55	55	55
	Black	Male	58	58	58	58	58	58	58	58	58	58	58	58
		Female	23	23	23	23	23	23	23	23	23	23	23	23
	White	Male	59	59	59	59	59	59	59	59	59	59	59	59
		Female	15	15	15	15	15	15	15	15	15	15	15	15
$\bar{X}$	Brown	Male	101	98.4	101	110	43.7	28.4	34.3	46.4	54.8	13.2	47.3	21.3
		Female	102	99.3	102	113	46.3	28.9	35.2	49.2	52.7	13.4	48.3	22.2
	Grey	Male	103	100	102	113	45.3	28.9	35.5	47.9	54.8	13.5	49	22
		Female	103	98.5	101	113	45.7	28.7	34.6	48.4	52.9	13.1	48.1	22.1
	Black	Male	103	101	103	113	45.6	29.4	35.4	49.3	55.1	13.6	49.2	22.8
		Female	102	99.4	102	112	45.9	28.1	34	48.5	53.5	13.2	48.7	22
	White	Male	109	106	108	115	47.7	30.6	36.8	52.6	58.5	13.8	49.7	21.8
		Female	104	100	103	114	45.1	28.4	35.1	49.3	55.2	13.5	48.6	22.2
	General		103.8	100.7	103.1	112.8	45.7	29.1	35.3	49	55	13.5	49	22
$s_{\bar{x}}$	Brown	Male	1.18	0.88	0.79	1.27	0.65	0.5	0.52	1.5	0.71	0.15	0.58	0.32
		Female	1.07	0.64	0.76	1.02	0.62	0.56	0.43	1.75	0.58	0.14	0.53	0.25
	Grey	Male	0.73	0.61	0.57	0.76	0.42	0.34	0.43	0.86	0.43	0.11	0.37	0.18
		Female	0.72	0.63	0.63	1.07	0.45	0.4	0.35	1.09	0.47	0.13	0.4	0.25
	Black	Male	0.91	0.81	0.76	0.72	0.54	0.36	0.49	1.03	0.69	0.13	0.39	0.28
		Female	1.12	1.09	0.95	1.72	0.76	0.74	0.63	1.74	0.67	0.25	0.68	0.38
	White	Male	1.19	1.15	1.09	0.93	0.66	0.57	0.5	1.05	0.75	0.16	0.39	0.24
		Female	2	1.96	1.88	1.8	0.75	0.69	0.89	1.89	1.53	0.25	0.58	0.44
s	Brown	Male	7.81	5.86	5.25	8.42	4.3	3.34	3.41	9.92	4.73	1	3.84	2.09
		Female	5.45	3.29	3.85	5.19	3.15	2.88	2.19	8.92	2.96	0.7	2.68	1.29
	Grey	Male	6.95	5.84	5.39	7.28	4.01	3.27	4.11	8.18	4.06	1	3.55	1.73
		Female	5.35	4.65	4.66	7.9	3.35	2.95	2.63	8.09	3.52	0.95	2.95	1.87
	Black	Male	6.96	6.17	5.75	5.46	4.08	2.73	3.73	7.88	5.23	0.98	2.95	2.09
		Female	5.36	5.24	4.55	8.26	3.65	3.57	3	8.36	3.19	1.2	3.25	1.83
	White	Male	9.12	8.83	8.35	7.14	5.1	4.39	3.86	8.1	5.77	1.21	3.01	1.82
		Female	7.75	7.6	7.27	6.98	2.9	2.67	3.44	7.34	5.91	0.97	2.26	1.7

BL, Body length (cm); WH, withers height (cm); HR, height at rump (cm); CC, chest circumference (cm); CD, chest depth (cm); CW, chest width (cm); HW, haunch width; TL, tail length (cm); HL, head length (cm); CAC, front shank circumference (cm); LL, limb length (cm); EL, ear length (cm); $\bar{X}$, Mean; s, Standard deviation; $s_{\bar{x}}$, Standard error.

**Table 2 animals-13-02366-t002:** Correlation coefficients between morphological features.

	BL	WH	HR	CC	CD	CW	HW	TL	HL	CAC	LL	EL
BL	1	0.762 **	0.767 **	0.614 **	0.541 **	0.642 **	0.584 **	0.444 **	0.581 **	0.515 **	0.532 **	0.233 **
WH	0.762 **	1	0.951 **	0.569 **	0.691 **	0.572 **	0.555 **	0.457 **	0.783 **	0.528 **	0.547 **	0.210 **
HR	0.767 **	0.951 **	1	0.586 **	0.649 **	0.590 **	0.571 **	0.444 **	0.755 **	0.533 **	0.551 **	0.240 **
CC	0.614 **	0.569 **	0.586 **	1	0.505 **	0.652 **	0.604 **	0.251 **	0.345 **	0.528 **	0.505 **	0.361 **
CD	0.541 **	0.691 **	0.649 **	0.505 **	1	0.378 **	0.337 **	0.186 **	0.096	0.344 **	0.432 **	0.280 **
CW	0.642 **	0.572 **	0.590 **	0.652 **	0.378 **	1	0.661 **	0.392 **	0.460 **	0.489 **	0.497 **	0.244 **
HW	0.584 **	0.555 **	0.571 **	0.604 **	0.337 **	0.661 **	1	0.438 **	0.472 **	0.411 **	0.481 **	0.333 **
TL	0.444 **	0.457 **	0.444 **	0.251 **	0.186 **	0.392 **	0.438 **	1	0.469 **	0.244 **	0.375 **	0.126 *
HL	0.581 **	0.783 **	0.755 **	0.345 **	0.096	0.460 **	0.472 **	0.469 **	1	0.434 **	0.380 **	0.045
CAC	0.515 **	0.528 **	0.533 **	0.528 **	0.344 **	0.489 **	0.411 **	0.244 **	0.434 **	1	0.562 **	0.291 **
LL	0.532 **	0.547 **	0.551 **	0.505 **	0.432 **	0.497 **	0.481 **	0.375 **	0.380 **	0.562 **	1	0.312 **
EL	0.233 **	0.210 **	0.240 **	0.361 **	0.280 **	0.244 **	0.333 **	0.126*	0.045	0.291 **	0.312 **	1

* (*p* < 0.05), ** (*p* < 0.01).

**Table 3 animals-13-02366-t003:** Importance of predictors in RF.

Inc Node	Purity
Colour	206.65191
Province	2418.43807
Sex	26.98572
Age	148.93556
WH	2147.68648
HR	2058.41290
CC	1286.96548
CD	611.77061
CW	1087.82828
HW	1102.90156
TL	497.52907
HL	1364.95827
CAC	510.62463
LL	561.35982
EL	261.88463

**Table 4 animals-13-02366-t004:** Coefficients determined for the body length target variable.

Model Term	Coefficient	*p*-Value	Importance
Intercept	23.554	0.000	
Province = 0	5.964	0.000	0.415
Province = 1	5.988	0.000	0.415
Province = 2	1.231	0.000	0.415
Province = 3	3.000	0.000	0.415
HR	0.425	0.000	0.231
CC	0.260	0.001	0.171
HL	0.260	0.001	0.065
HW	0.289	0.003	0.051
TL	0.072	0.015	0.034
CAC	0.635	0.017	0.033

This coefficient was set to zero as it was redundant.

**Table 5 animals-13-02366-t005:** Cook’s distance values for body length (BL).

Sample ID	Value	Cook’s Distance
323	97	0.043
219	97	0.041
28	126	0.034
209	97	0.032
45	123	0.028
50	102	0.022
55	95	0.021
292	124	0.020
282	117	0.016
303	117	0.016
54	108	0.015
3	97	0.012
240	95	0.012

**Table 6 animals-13-02366-t006:** Results of MARS algorithm regarding the prediction of body length in donkeys.

Variables	Coefficients
(Intercept)	105.936153
h(108-HR)	−1.063314
h(HR-108)	0.286334
h(CC-115)	4.494783
h(CC-116)	−8.182082
h(118-CC)	0.977991
h(CC-118)	13.999379
h(CC-119)	−7.825563
h(41-HW)	−1.210453
h(HW-41)	−1.296884
h(51-TL)	−0.138144
Province Mardin * h(WH-106)	0.443792
h(98-HR) * h(118-CC)	−0.065563
h(108-HR) * h(36-HW)	0.144160
h(108-HR) * h(CAC-13.5)	0.906941
h(108-HR) * h(LL-45)	0.055649
h(108-HR) * h(45-LL)	0.048355
h(CC-113) * h(41-HW)	0.908293
h(114-CC) * h(41-HW)	−0.111693
h(CC-114) * h(41-HW)	−1.079222
h(CC-118) * h(TL-42)	0.116470
h(CC-118) * h(42-TL)	0.304491
h(41-HW) * h(HL-59)	0.211118
h(41-HW) * h(59-HL)	0.061253
Province Kars * h(41-HW) * h(59-HL)	−0.052772
h(2-Age) * h(41-HW) * h(59-HL)	−0.049791
h(108-HR) * h(CW-25) * h(13.5-CAC)	0.067626
h(108-HR) * h(25-CW) * h(13.5-CAC)	0.088635
h(108-HR) * h(HW-36) * h(CAC-13)	0.534718
h(108-HR) * h(HW-36) * h(13-CAC)	0.556636
h(108-HR) * h(46-TL) * h(CAC-13.5)	7.388817
h(CC-113) * h(41-HW) * h(46-LL)	−0.312434
h(108-HR) * CC * h(46-TL) * h(CAC-13.5)	−0.066624
h(108-HR) * h(CC-111) * h(HW-36) * h(LL-48)	−0.016711

## Data Availability

The data presented in this study are available in Appendix A.

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
