# Peer review of "Investigation of the Relationships between Coat Colour, Sex, and Morphological Characteristics in Donkeys Using Data Mining Algorithms"

_animals, 2023, doi:10.3390/ani13142366_

Round 1

Reviewer 1 Report

I consider the paper to not be publishable in the current form due to the issues I will present below. If the authors were to make edits that fulfill the expectations, I would recommend acceptance of the paper:

First, methods are not described at length. Other than describing a couple details about each adopted algorithm, it is important to explain the whole data processing pipeline, including especially any data pre-processing technique, segmentation (if applicable, given the time component that may be present in the dataset), and most of all the adopted methodology to split the dataset into training and testing. 
Besides, it is unclear at which point in time the set of features is collected for each animal (at birth?) and at which point in time the prediction is attempted to be made. It seems like maybe the authors use information about the donkey at birth to figure out what the weight is going to be as an adult, but it is not clearly stated. 

Second, several studies are mentioned in the introduction, as well as in the discussion, but it is not exactly clear what is the contribution of this work with respect to the state of the art. The authors do not highlight any flaw in these studies, and they do not highlight what the contribution of this work really is. Just adopting other machine learning algorithms is not enough of a contributions, since these algorithms are well known and not designed by the authors themselves. Adopting just another algorithm is not a contribution. They do carry out somewhat of an analysis of the most important features for this type of predictions, but there should be an emphasis on this to explicitly clarify the contribution of this work. 

Other details:

the abstract highlights some measurements, but it is not clear if these are mean measurements across the data they have or what else do they represent.

Some of the plots are unreadable, especially in printed form. Besides, other results, such as Table 6 or the supplementary material really are badly displayed. You can't expect a reader to just go through each line of a table that is longer than a full page. Results need to be displayed and summarized in an efficient and compact matter that clearly state what are the results. Please, use SVG for the plots, or, at minimum, higher pixel density.

As mentioned before, the discussion highlights a few similar works. The authors limit the discussion to simply mentioning these works and saying how some are similar, some are different. So what? How is this relevant with respect to this work? Performing a similar study on just another dataset is no contribution at all. It needs to be emphasized how this work is advancing the state of the art. 

Writing. I do not fully appreciate the fact that paragraphs are so short in the introduction. It seems like there is almost one sentence per paragraph.

Author Response

Dear Reviewer,

This document is a response letter on the manuscript # animals-2438262 entitled "Investigation of the relationships between coat colour, sex, and morphological characteristics in donkeys using data mining algorithms".

Dear Editor,

Many thanks for sharing valuable comments of you and reviewers with us on improving the manuscript # animals-2438262 entitled "Investigation of the relationships between coat colour, sex, and morphological characteristics in donkeys using data mining algorithms". We are happy that the manuscript will be acceptable for evaluation in "Animals". We have given answers to all comments of the reviewers evaluating our manuscript using red and blue color fonts. Also, red (Reviewer 1) and blue (Reviewer 2) color font on the revised manuscript has been used for indicating all the corrections made by Reviewers.

With Best Regards

Assoc. Prof. Åženol Çelik

Reviewer 2 Report

This study, and its methodology, are out of my exact expertise, so this review may or may not be helpful. It is somewhat difficult to follow the exact protocol that was used. One important issue is whether or not the different geographic origins of the donkeys have a significant effect on their size. This should be included as a variable. The classification of color is at least a little bit weak, because it is difficult to tell exactly what "brown" is, or what "grey" is. Some of the individual color variants in donkeys (such as chestnut) appear to have effects on size, so accurate color classification is important in a study like this.

The English is good, and clearly understandable.

Author Response

Dear Reviewer,

This document is a response letter on the manuscript # animals-2438262 entitled "Investigation of the relationships between coat colour, sex, and morphological characteristics in donkeys using data mining algorithms".

Dear Editor,

Dear Reviewer,

Many thanks for sharing valuable comments of you and reviewers with us on improving the manuscript # animals-2438262 entitled "Investigation of the relationships between coat colour, sex, and morphological characteristics in donkeys using data mining algorithms". We are happy that the manuscript will be acceptable for evaluation in "Animals". We have given answers to all comments of the reviewers evaluating our manuscript using red and blue color fonts. Also, red (Reviewer 1) and blue (Reviewer 2) color font on the revised manuscript has been used for indicating all the corrections made by Reviewers.

With Best Regards

Assoc. Prof. Åženol Çelik

Round 2

Reviewer 2 Report

This study is more related to techniques of data mining than to the characteristics of the donkeys. As such, it will appeal to only a small proportion of readers. Despite this drawback, it does appear to be comprehensive and accurate. I suspect that it has minimal application to most readers, but that might be mistaken.

The English is good, and is understandable despite a few relatively minor lapses.